# Severe Hypoxia Does Not Offset the Benefits of Exercise on Cognitive Function in Sedentary Young Women

**DOI:** 10.3390/ijerph16061003

**Published:** 2019-03-20

**Authors:** On-Kei Lei, Zhaowei Kong, Paul D. Loprinzi, Qingde Shi, Shengyan Sun, Liye Zou, Yang Hu, Jinlei Nie

**Affiliations:** 1Faculty of Education, University of Macau, Macao, China; mb54871@connect.um.edu.mo (O.-K.L.); zwkong@um.edu.mo (Z.K.); 2Department of Health, Exercise Science and Recreation Management School of Applied Sciences, The University of Mississippi, Oxford, MS 36877, USA; pdloprin@olemiss.edu; 3School of Physical Education and Sports, Macao Polytechnic Institute, Macao, China; qdshi@ipm.edu.mo (Q.S.); jnie@ipm.edu.mo (J.N.); 4Institute of Physical Education, Huzhou University, Huzhou 313000, China; sysun@zjhu.edu.cn; 5Lifestyle (Mind-Body Movement) Research Center, College of Sports Science, Shenzhen University, Shenzhen 518060, China; 6Sports Science Research Center, Beijing Sport University, Beijing 100084, China; huyang@bsu.edu.cn

**Keywords:** hypoxia, exercise, reaction time, accuracy, peripheral oxygen saturation

## Abstract

*Purpose:* To examine the effect of acute moderate-intensity continuous exercise performed under normobaric severe hypoxia on cognition, compared to sea-level normoxia. *Methods:* Thirty healthy inactive women randomly performed two experimental trials separated by at least three days but at approximately the same time of day. Executive functions were measured during the follicular stage via an interference control task before (rest) and during exercise with 45% peak power output under normobaric normoxia (PIO_2_ = 150 mmHg, FIO_2_ = 0.21), and (2) hypoxia (PIO_2_ = 87 mmHg, FIO_2_ = 0.12, simulated at an altitude of 4000 m). Reaction time (RT), accuracy rate (AC), heart rate, ratings of perceived exertion, and peripheral oxygen saturation (SpO_2_) were collected before and during exercise. *Results:* RT (*p* < 0.05, η^2^_p_ = 0.203) decreased during moderate exercise when compared at rest, while a short bout of severe hypoxia improved RT (*p* < 0.05, η^2^_p_ = 0.134). Exercise and hypoxia had no effects on AC (*p* > 0.05). No significant associations were found between the changes of RT and SpO_2_ under the conditions of normoxia and hypoxia (*p* > 0.05). *Conclusions:* At the same phase of the menstrual cycle, a short bout of severe hypoxia simulated at 4000 m altitude caused no impairment at rest. RT during moderate exercise ameliorated in normoxia and severe hypoxia, suggesting that both exercise and short-term severe hypoxia have benefits on cognitive function in sedentary young women.

## 1. Introduction

A number of studies have shown that exercise, especially moderate-intensity exercise, not only increases cardio-metabolic health [1,2], but also improves mental health [2,3,4] in young [5,6] and aged populations [7,8,9]. A single bout of moderate exercise enhances cognitive function through increasing arousal level and improving goal oriented processing in the brain [10]. Additional mechanisms include exercise-induced increases in neurotransmitters, such as serotonin [11], neuronal growth factors [12,13,14], as well as activation of prefrontal cortex activity [15].

More recently, an emerging line of research has shown that, apart from the benefits of improving cardiorespiratory fitness in sedentary populations, hypoxic exercise seems to be a novel treatment strategy for weight loss and comorbidities in obese subjects [16,17,18,19,20]. However, hypoxia itself, may induce negative cognitive-related consequences as severity increases [4,5]. Insufficient oxygen is delivered to the brain during exercise under hypoxia, therefore leading to inadequate cerebral oxygenation and cerebral blood flow (CBF) [21]. As pulse oximetry saturation (SpO_2_) and cerebral oxygenation react similarly in response to hypoxia, SpO_2_ is generally used to indicate hypoxia degree and can also be utilized as an important predictor for cognitive function under hypoxia [22]. During moderate exercise under severe hypoxia, SpO_2_ decrement has been reported to attenuate exercise-induced cognitive improvement and increase reaction time (RT) in a Go/NoGo task in males [5]. Furthermore, there is a moderate negative correlation between the changes of SpO_2_ and RT [5], suggesting that hypoxia, especially severe hypoxia, may offset the benefits of exercise on cognition. Nevertheless, the findings regarding the cognitive responses to moderate exercise under severe hypoxia are inconsistent in the existing literature [5,23,24]. Using similar Go/NoGo tasks and the same fraction of inspired oxygen (FIO_2_ = 0.12, simulating an altitude equivalent to 4000 m), either unaffected [25] or improved [4,5,24,26] cognitive performance in response to exercise has been reported. The discrepant results may be caused by differences in exercise intensities, subjects’ demographic characteristics (e.g., sex, physical fitness level and health condition), the timing for cognitive task administration and task difficulty [4,5,23,24,27,28]. Based on the benefits caused by hypoxic exercise on physical health [1,14,17,20], it is indispensable to clarify whether exercise under severe hypoxia has any adverse effects on cognitive function.

Complex cognitive processes, such as executive control (behavioral responses to facilitate goal attainment), error processing (a component of cognitive control involved in processing information to prevent and/or correct errors), and inhibitory control (ability to inhibit or regulate prepotent attentional or behavioral responses), are partially regulated by the prefrontal cortex (PFC) [29,30], which is a brain structure likely to be activated when participants performed the Go/NoGo task [25,26,31,32,33,34]. Previous research aimed at investigating the neural underpinnings of Go and NoGo performance demonstrated, via functional magnetic resonance imaging (fMRI), that PFC was associated with the inhibitory component (NoGo) of this task, implicating that the Go/NoGo task is involved in response inhibition RT [35]. 

The changes in cognitive performance at various levels of hypoxia are well documented in men [4,5,24,27,36] and in a mixed sample of men and women [23,28]. However, very few related studies have been conducted on women [37]. Numerous studies investigating sex difference in blood oxygen saturation reported that females have higher average of SpO_2_ than males [38] due to sex-specific differences in hormones [39], which regulate breathing control and indirectly induce changes in function of the respiratory system [37,38]. Moreover, it has been acknowledged that estrogen has beneficial effects on CBF [39] as increasing estrogen levels can reduce cerebrovascular resistance [40] and proliferate CBF velocity [41]. Given the increased cerebral artery vasodilation stimulated by the increase of estrogen, women are reported to have greater basal CBF than men under hypoxia, especially in the early follicular stage [42,43,44]. Considering the neuroprotective role of estrogen in response to hypoxia [45] and a greater resistance to hypoxia in females [46], the present findings among male-dominant studies may not be applicable to women. 

As such, it is critical that future studies using female subjects be conducted when evaluating the interrelationships among acute exercise, hypoxia and cognition. Importantly, such studies need to carefully consider the reproductive phase of the woman, as phase-specific differences in gonadal steroids has also been demonstrated to influence various cognitive processes, such as executive functions [39]. For example, increased emotional memory has been observed in the phase of the menstrual cycle with higher progesterone levels [39]. Moreover, follicle-stimulating hormone has been found to be negatively, whereas luteinizing hormone positively, correlated to visuospatial ability [47], and positive correlations between estradiol levels and paired-association learning have also been documented [48]. 

In addition to the well-established moderate exercise benefits [1,2,3,4,5,6,7,8,9,10] and hypoxia-related cognitive risks [49,50,51,52], the combined effects of hypoxia and moderate exercise on cognitive performance specifically in young inactive women may provide a valuable insight in identifying the physiological factors that affect cognition. In this study, using a relatively low hypoxia (FIO_2_ = 0.12), we aimed to examine the acute response of hypoxic exercise on cognition in females during the same menstrual cycle when habitual physical activity and diet were controlled, as both of these parameters may influence cognitive function [39,53]. We hypothesized that severe hypoxia would impair cognitive performance, and the benefit resulting from moderate exercise would be offset when the hypoxia was imposed to the sedentary young women.

## 2. Methods

### 2.1. Study Participants

Volunteers were publicly recruited via flyers posted on the campus of the University of Macau. The sedentary, healthy young women were included if they met the following criteria: (1) residence at altitude below 1300 m; (2) neither previous experience of hypoxic training nor prior engagement in any regular exercise; (3) non-smoking and alcohol drinking habits; (4) having a self-reported regular menstrual cycle with 28–34 days of length; (5) and not taking any form of the contraceptive pill or other drugs. After the screening phase, 30 eligible subjects (age: 22.6 ± 3.2 years; body mass index: 22.1 ± 3.1 kg·m^−2^; VO_2_peak: 26.3 ± 5.0 mL·kg^−1^·min^−1^) from 54 volunteers were recruited to participate in this study. All participants gave their informed consent for inclusion before they participated in the study. The protocol was approved by the Ethics Committee of the University of Macau (MYRG2018-00216-FED). Participants provided consent per the principles of the Declaration of Helsinki. All participants completed the experimental sessions during their follicular stage, which was defined as the time from the first day to the seventh day after the menses, to ensure a stable physiological state. 

### 2.2. Preliminary Testing and Familiarization 

The preliminary testing session was performed at the beginning of the follicular stage of the menstrual cycle. Body weight was measured with light clothing and bare feet in a fasting state (Tanita MC-180 MA, Tanita Corporation, Tokyo, Japan). Participants exercised on a stationary cycle ergometer (Monark 839e, Vansbro, Sweden) with a graded workload that increased each stage by 15 watts every 3 min beginning at 25 watts at 60 revolutions per minute (rpm). VO_2peak_ was examined using a pre-calibrated breath-by-breath analysis system (Meta-Max 3B, Cortex Biophysik GmbH, Leipzig, Germany). Heart rate (HR; Polar F4M BLK, Kempele, Finland) and ratings of perceived exertion (RPE; 6–20 Borg scale) were recorded during the graded maximal exercise test. The test was terminated when participants performed to volitional exhaustion, were not able to maintain a pedaling rate of 60 rpm any longer, or RPE reached 18 or higher. The highest VO_2_ averaged over the final 30 s was identified as the VO_2peak_. During the familiarization, subjects learned the experimental procedures and practiced the cognition task multiple times until the number of erroneous trials in each session were less than five (one practice session includes 40 trials). 

### 2.3. Experimental Procedure

After a session of preliminary testing and familiarization with the executive cognition task, all subjects visited the laboratory twice at their follicular stage. They performed two experimental trials separated by at least three days but at approximately the same time of day: normoxia (NOR, inspiratory oxygen pressure (PIO_2_) = 150 mmHg, FIO_2_ = 0.21) and normobaric hypoxia (HYP, PIO_2_ = 87 mmHg, FIO_2_ = 0.12). We adopted a single-blinded crossover design, with the subjects being unaware of the oxygen condition, and the two experimental trials were performed in a counterbalance order to offset any potential learning effects. Hypoxic training system (Everest Summit II Hypoxic Generator, New York, NY, USA) was used to monitor the conditions for normobaric hypoxia trial corresponding to ~4000 m. At the beginning of the experiment, participants rested quietly in a seated position for 10 min with a mask connected to either normoxia or hypoxia condition in which they were blinded to. Then, they cycled for 10 min at a free pedaling rate at 45% of peak power output (PPO), which was calculated with the formula: PPO = Wcom + (t/180) × ΔW, where Wcom is the workload of the last stage in the VO_2peak_ test, t is the number of seconds for which the incomplete stage was sustained, and ΔW is the load increment (25W) [54]. Cognitive tasks were performed on the 10th minute (at rest) and on the 17th minute (i.e., 5 min after exercise started) of the experimental timeline. The whole experimental session lasted for approximately 22 min (Figure 1). In addition, SpO_2_ was measured by a pulse oximeter (Radical-7 Pulse CO-Oximeter, Masimo, Irvine, CA, USA) placed on the right index finger [55]. HR and RPE were recorded before and after each cognitive task. The diameter of the resistance vessels in the brain is much greater in severe hypoxia equivalent to altitudes above 4500 m than in normoxia [56], implying that this may be the threshold in impairing cognitive function [57]. Experimental procedures are presented in Figure 1. 

### 2.4. The Interference Control Task

In this study, cognitive performance was evaluated through a modified version of the traditional Go/NoGo task [4,5], herein referred to as an interference control task, which provides a simple computerized test to assess these PFC-dependent cognitive processes. The traditional Go/NoGo task involves a cognitive and physical inhibitory component during the NoGo trial. For our experiment, we modified this component by asking participants, during the NoGo trial, to press a different key button than that required for the Go trial. Further details are explained below. The impetus for this modification is because far fewer studies have evaluated the effects of exercise on this [interference control] executive function component. 

The RT and accuracy rate (AC) were collected and analyzed using an E-prime program which was installed on a 15’6 inch Lenovo B560 laptop. The laptop was placed on a portable desk in front of the cycle ergometer that can be easily moved to subjects for the cognitive task during exercise. One cognitive task included 40 trials and took approximately 2 min to complete. For each stimulus, two symbols would take turn to show on the middle of the screen (i.e., a square printed in red or blue color, and then followed by a number or letter printed in black color). A combination of a red square followed by a number or a combination of a blue square followed by a letter indicated the “Go” signal, in which subjects had to respond by pressing the “F” button on the keyboard with the left index finger. If the stimulus appeared in a contrary combination of a red square followed by a letter or blue square followed by a number, “NoGo” signal was presented. Subjects needed to respond by pressing the “J” button with the right index finger. The stimulus disappeared from the screen when a response was given, or else remained on screen for 2 s. A fixation cross would appear on screen for 2 s after each trial as an inter-stimulus interval. The chances of responding “Go” and “NoGo” are fifty percent, respectively. In this study, the average response time (ms) and the AC (%) form all trials were calculated to measure inhibitory control. 

### 2.5. Physical Activity and Dietary Assessments

In order to minimize the potential confounding effects of changes in energy balance, physical activity and dietary behavior, participants were required to strictly maintain their habitual dietary and lifestyle behavior during the experiment. Physical activity and dietary behavior were recorded for 2 days (one day before and the day of the experiment) during each experimental trial. Physical activities were monitored using a previously validated pedometer (Ymax SW-200 Digiwalker, Tokyo, Japan) [1,58], while the size and weight of food and beverage intakes were recorded by the Chinese nutrition action program (CNAP) questionnaire [1,59]. The macronutrient proportion and daily energy intake were calculated using the Sports Nutrition Centre of the National Research Institute of Sports Medicine (NRISM) dietary and nutritional analysis system (version 3.1).

### 2.6. Statistical Analysis

After the normality and homogeneity of the variance were determined, paired t-tests were conducted to confirm the consistency of physical activity and dietary behavior. Two-way repeated-measures ANOVAs were conducted for conditions of oxygen (NOR and HYP) and exercise (rest and exercise) as within-subject factors for the variables of physiological parameters (i.e., HR, RPE and SpO_2_) and the interference control task. The degrees of freedom were corrected using the Huynh Feldt Epsilon when the assumption of sphericity was violated. Tukey’s post hoc test was conducted to compare mean differences where appropriate. Partial eta-squared (η^2^_p_) was used to determine the effect sizes of the main and interaction effects. The effect size was considered small if η^2^_p_ < 0.06 and large if η^2^_p_ > 0.14 [60]. In addition, Pearson’s product-moment correlation coefficients were calculated to evaluate the correlations between the changes of RT and SpO_2_. All data is expressed as mean ± SD. The significance level was set at 0.05 (IBM SPSS Statistics Base 22.0, IBM, Chicago, IL, USA). 

## 3. Results

### 3.1. Habitual Physical Activity and Dietary Profile

There were no significant differences in daily activity (NOR: 8614 ± 2782 steps; HYP: 8109 ± 2797 steps) between the two trials. With the similar composition of nutrients by 51% carbohydrate, 16% protein and 33% fat, no significant trial differences were found in calorie intake (NOR:1780 ± 429 kcal; HYP:1712 ± 394 kcal).

### 3.2. Physiological Parameters

Table 1 displays the physiological responses to the two experimental trials. Condition (F(1,29) = 14.195, *p* < 0.001, η^2^_p_ = 0.348) and exercise (F(1,29) = 475.500, *p* < 0.001, η^2^_p_ = 0.942) had a significant main effect on HR change. When compared to normoxia and at rest, higher HR were found in hypoxia and during exercise. Regarding SpO_2_, condition (F(1,29) = 308.173, *p* < 0.001, η^2^_p_ = 0.904) and exercise (F(1,29) = 52.244, *p* < 0.001, η^2^_p_ = 0.634) had significant main effects and an interaction effect (F(1,29) = 40.930, *p* < 0.001, η^2^_p_ = 0.574). Further post-hoc Tukey analyses indicated that, when compared to normoxia, SpO_2_ in hypoxia experienced significant decreases either at rest (*p* < 0.001) or during exercise (*p* < 0.001). The reduction of SpO_2_ during exercise (*p* < 0.001) was only found in hypoxia as compared in normoxia. Exercise (F(1,29) = 184.415, *p* < 0.001, η^2^_p_ = 0.864) but not hypoxia (F(1,29) = 1.710, *p* = 0.201, η^2^_p_ = 0.056) caused an increase in RPE.

### 3.3. Cognitive Function

Figure 2 shows the results of RT (Figure 2A) and AC (Figure 2B) in the interference control task under different conditions. There were significant main effects of exercise (F(1,29) = 8.336, *p* = 0.011, η^2^_p_ = 0.203) and condition (F(1,29) = 5.425, *p* = 0.043, η^2^_p_ = 0.134) on RT, whereas no interaction effect was found between exercise and condition (F(1,29) = 0.524, *p* = 0.573, η^2^_p_ = 0.011) on RT. Based on the interest in knowing the difference of condition on RT at rest, we did a further paired *t*-test analysis and it demonstrated that resting RT under hypoxia was faster than under normoxia (*p* = 0.012)). These results indicated that both moderate exercise and short-term hypoxia improved RT. Regarding the AC of the interference control task, no main effects of exercise (F(1,29) = 0.487, *p* = 0.445, η^2^_p_ = 0.020) or condition (F(1,29) = 0.777, *p* = 0.796, η^2^_p_ = 0.002) was found, and there was no interaction effect (F(1,29) = 0.030, *p* = 0.798, η^2^_p_ = 0.002) neither. The changes of RT were not associated with the changes of SpO_2_ in normoxic (*r* = 0.16, *p* = 0.416) and hypoxic conditions (*r* = −0.20, *p* = 0.309).

## 4. Discussion

The purpose of this study was to examine the effect of acute moderate-intensity continuous exercise performed under normobaric severe hypoxia on cognition, compared to sea-level normoxia, among young sedentary women. The key findings in this study were that both acute moderate-intensity exercise and severe hypoxia (~30 min) improved RT, with these alterations not being associated with changes in SpO_2_.

### 4.1. Cognitive Function at Rest Under Hypoxia

In the present study, at the beginning of each experimental trial, we measured cognitive function at rest under normoxia and hypoxia. In contrast to our hypothesis, RT decreased with an unaffected AC after resting under hypoxia for 10 min. This result is consistent with some previous findings, suggesting that hypoxia did not impair cognitive performance [4,5]. However, using similar hypoxic exposure levels (FIO_2_ = 0.12 or 0.125), there are several studies reporting unchanged [28] or slower RT [23], and decreased AC [24] in male subjects [24] and mixed subjects [23,28]. Compared to these studies [23,28], the female subjects in the present study showed a relatively higher SpO_2_ value. The different SpO_2_ response to the same hypoxia level between males and females may, in part, be explained by the stronger resistance to hypoxia in women [46]. This may have occurred given the fact that we conducted this study during the early follicular stage of the female subjects when estrogen hormones were at a relatively high level, and thus, may provide a neuroprotective response to hypoxia [45]. In support of this notion, various cognitive parameters are heightened during the follicular stage, at times of higher estrogen or progesterone, were reported in previous studies [39,48,61]. 

### 4.2. Cognitive Function During Exercise Under Hypoxia

Studies regarding the impact of hypoxic exercise on cognitive performance are mainly focused on males [4,5,24,27,36] or mixed subjects [23,28]. Thus, studies on this topic using female subjects is rare. The present study revealed that acute moderate exercise increased the reaction speed (η^2^_p_ = 0.203, indicating large effect) without affecting the response accuracy in sedentary women, suggesting that moderate exercise under severe hypoxia promoted a beneficial effect on cognitive function. In the present study, SpO_2_ was reduced to 87 ± 6% after resting in hypoxia and further decreased to 77 ± 7% after exercise in hypoxia. However, no adverse cognitive consequence was observed, which might be compensated by an increased CBF. It has been reported that when a conflict exists between preserving brain O_2_ delivery or restraining CBF to avoid potential damage and an elevated perfusion pressure after exercise, the priority is given to brain O_2_ delivery to secure maintenance of cognitive function and avoiding potential damage by hypoperfusion [56,62]. Furthermore, the observation of that severe hypoxia did not attenuate exercise-induced benefits in cognition in young women in the present study is supported by, at least partially, the aforementioned fact that women have fundamentally higher basal CBF than men under both normoxic [63,64] and hypoxic conditions [65] because of the increased cerebral artery vasodilation stimulated by the increase of estrogen [42,43,44,65]. Despite this, further studies should consider evaluating whether the follicular stage moderates the interrelationship between exercise and hypoxia on cognitive function in females. 

Using a similar experimental design and the same hypoxic condition, similar cognitive results were previously reported in male subjects [5], despite the fact that they used a relatively lower exercise intensity. Under normoxia and hypoxia, the present study used the same absolute intensity (45% PPO), which was equivalent to 73% HRmax under normoxia and 80% HRmax under hypoxia. Differently, Komiyama et al. [5] evaluated Go/NoGo task performance during nomoxic and hypoxic exercise with a relative intensity at 50% VO_2peak_ (corresponding to ~60% of HRmax), which was lower than our study. Despite the greater exercise intensity, the reduction of SpO_2_ in the female subjects of our study was the same as the male subjects in their study [5], both decreasing from 87% to 77%. The lack of difference in SpO_2_ response under higher-intensity exercise might be explained by the fact that females have higher resistance to hypoxia [46].

Although interference control tasks can be used to evaluate motor executive and inhibitory processing [28], conclusion of the effect of exercise under short-term normobaric hypoxia on cognition still remains uncertain due to diverse methodological issues (e.g., differences in cognitive task-specific measure, subjects, severity of hypoxia, hypoxic exposure time) [5,23,24,27,28,36]. The high AC in our study (NOR: 89%, HYP: 90%) indicated that cognitive data was not affected by subjective responses, such as skill level and motivation. The task difficulty and testing time might have been too low to cause a negative effect on cognitive performance, which might account for the discordance between conclusions of cognitive impairment resulting from hypoxic exercise in some previous studies [23,28] and cognitive enhancement. Our results suggest that the hypoxic condition (12% O_2_) was not sufficient to affect cognitive performance. In addition, the present study investigated only one cognitive task, thus, future investigations should consider evaluating multiple tasks when designing programs to better clarify the relationship between acute exercise and cognition under different levels of hypoxia.

### 4.3. Strengths and Limitations 

This study had several strengths, including the relatively large sample size, single-blinded oxygen conditions, cognition assessment in the same phase of the menstrual cycle, and a monitored status of habitual physical activity and energy intake, which the majority of others studies having not considered [5,23,24,26,45,46,66]. Despite these strengths, there are several limitations of our study. First, cerebral blood flow and cerebral oxygenation was not recorded, and thus, mechanism regarding cognitive processes under hypoxic exercise needs to be further evaluated. Additionally, calendar-based measurement of menstrual cycle alone may be insufficient for interpretation, and the hormones of estrogen and progesterone should be tested to confirm the same phase of the menstrual cycle. 

## 5. Conclusions

In summary, the present study found that a short bout (~30 min) of severe hypoxia improved cognitive function when compared to normoxia in young sedentary women with the same menstrual period, similar daily energy intake and habitual physical activity. Furthermore, a short bout of exercise induced cognitive benefits in RT, and these alterations were not associated with the changes in SpO_2_. In addressing the discrepant findings and accounting for differences in cognitive function, both absolute and relative exercise intensities should be considered in future research. As males and females seem to have different physiological and cognitive responses to severe hypoxia, whether sex-specific hormones influence this process needs to be further clarified. Moreover, future studies may also consider adopting different exercise modes (e.g., high-intensity interval exercise, resistance exercise) and using more sophisticated and higher-order cognitive tasks to evaluate cognitive function, meanwhile, monitoring CBF and cerebral oxygenation profiles to reveal the potential mechanisms.

## Figures and Tables

**Figure 1 ijerph-16-01003-f001:**
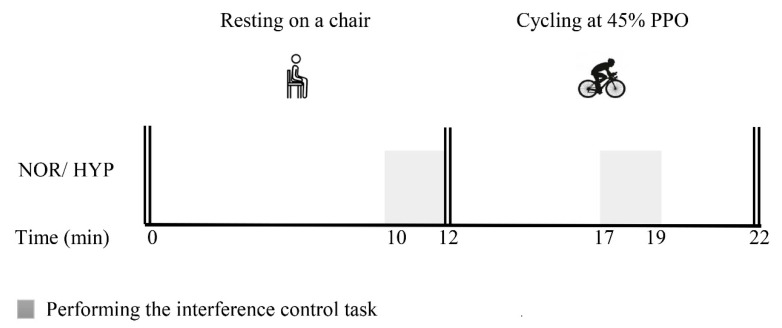
Schematic diagram of study protocol. PPO, peak power output; NOR, normoxia; HYP, hypoxia.

**Figure 2 ijerph-16-01003-f002:**
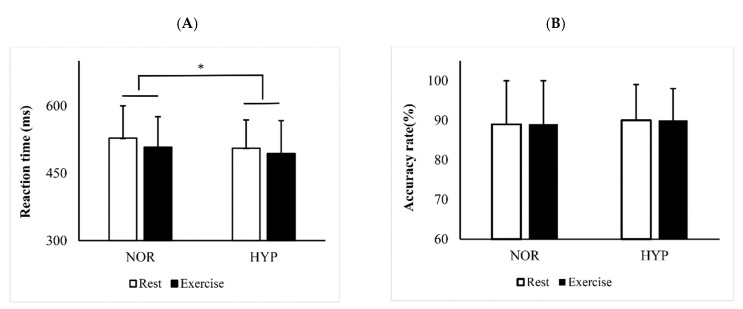
RT (**A**) and AC (**B**) in the interference control task under normoxia (NOR) and hypoxia (HYP). RT, reaction time; AC, accuracy rate. * *p* < 0.05 (NOR vs. HYP).

**Table 1 ijerph-16-01003-t001:** Physiological responses to moderate intensity exercise in normoxia or hypoxia.

Variables	NOR		HYP	
	Rest	Exercise	Rest	Exercise
HR	78 ± 11	129 ± 15 ^b^	84 ± 12	141 ± 18 ^ab^
%HR_max_		73 ± 8		80 ± 10 ^a^
RPE	7 ± 1	11 ± 2 ^b^	6 ± 1	12 ± 2 ^b^
SpO_2_	98 ± 2	97 ± 3	87 ± 6 ^a^	77 ± 7 ^ab^

NOR, normoxia; HYP, hypoxia; HR, heart rate; %HR_max,_ percentage of maximum heart rate; RPE, rate of perceived exertion; SpO_2_, pulse oximetry saturation; ^a^: *p* value less than 0.05 under HYP vs. NOR; ^b^: *p* value less than 0.01 under HYP and NOR.

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
