# Peer review of "Severe Hypoxia Does Not Offset the Benefits of Exercise on Cognitive Function in Sedentary Young Women"

_ijerph, 2019, doi:10.3390/ijerph16061003_

Round 1
Reviewer 1 Report
Manuscript: ijerph-438545
Title: Severe Hypoxia Does Not Offset the Benefits of Exercise On Cognitive Function in Sedentary Young Women
General comments
The authors investigated the acute effect of acute moderate-intensity continuous exercise performed under hypoxia on cognition, in young women. Considering that there is a gap in research focusing on these aspects, the study is timely. Another positive aspect of this study is the relatively large sample size for this type of investigation (n=30 all female). Previous studies included up to 30 participants, but the sample sex was often mixed or included just male subjects.
Shortcomings of this study include: i) the lack of a comprehensive rationale for research; ii) some gaps regarding the methods, especially regarding the description of the cognitive measure and the statistical analysis; iii) the lack of clarity in presenting some of the results. Partially due to the highlighted shortcomings, the discussion section appears somewhat limited. Please refer to the specific comments for further details.
Specific comments
Major
Introduction
Line 45: The authors should expand background on the relationship between hypoxia and cognitive functions. The current version is quite light and does not allow the reader to understand how previous studies (references 5, 15, 16 and 17) have explored the topic; what study design have they adopted; what sample age and size; what measures of cognition have been used. Additionally, some relevant articles have been overlooked (see, McMorris, T., Hale, B. J., Barwood, M., Costello, J., & Corbett, J. (2017). Effect of acute hypoxia on cognition: A systematic review and meta-regression analysis. Neuroscience & Biobehavioral Reviews, 74, 225-232 for review).
Line 49: the authors refer to previous study exploring hypoxia and cognition to have gotten to inconsistent findings, but the reader is left with no details on what these findings were. What did they found? Why is this inconsistent?
Lines 52-53: “Complex cognitive processes, such as executive control, error processing, and inhibitory control, are partially regulated by the prefrontal cortex (PFC)”. Please define executive control, error processing and inhibition control with references. Also please explain why it is important to specify that these are regulated by the prefrontal cortex – considering that your study is not trying to measure anything related to a specific region of the brain, it might be sufficient to specify what the cognitive processes of interest are.
Lines 53-54: “The Go/NoGo task provides a simple computerized test to assess these PFC-dependent cognitive processes”. This sentence seems to be better suited in the Methods section. Also, I think the authors should be more specific on what the GNG actually measures: previous studies highlighted that it mainly taps response inhibition and attention, but I am not sure the go/no-go task is an accurate measure for the overall executive control.
Line 61-63: In what way the “phase-specific differences in gonadal steroids has been shown to influence various cognitive processes, such as visual and spatial processing, and executive functions”? Please specify.
Methods
Line 101: please specify how the randomization was performed.
Lines 110-111: Please provide more details on the protocol of the cognitive task. Particularly please explain how participants completed the cognitive task during exercise. What was the setup? (also see major comment on Lines 122-124)
Line 112: why did you measure pulse oximetry saturation (SpO2)? There is no background on why you have included this measure.
Lines 119-120: Figure 1 needs some work: i) from the figure it is not clear how to locate Normoxia; ii) do the arrows represent the start and end of the GNG task? I suggest replacing them with a gray pattern. The arrows make it seem you have measured participants twice at rest and twice during exercise, which I suppose you did not do; iii) Please indicate "time (min)" under the X axis. Also, please note that the numbers under the figure do not follow a logical order. Indeed, they appear to be placed at random: the space between 10 and 12 is greater than the one between 0 and 10, and the one between 15 and 22; iv) the acronym PPO should be defined in the notes/legend.
Lines 122-124: the description of the cognitive task used to measure the main outcome of the study very limited and currently lacks important information: did you use the original paradigm developed by Donders in 1969 or is it a modified version? If it is a modified version what program did you use to modify the task? What parameters did you use? How many trials? How many blocks? How much time did each trial last? How much time between each trial? Did you have a fixation cross on the screen between trials? Did the participant have a practice block? How long did the practice go for? Did you use the ‘raw’ data for your analysis (i.e., response time to each individual trial) or made an average of them? Did you use the response time from go tasks or both go and no-go trials? How did you treat premature responses? Did you extract the accuracy rate from all trials or only the no-go trials? Did you use a laptop or a desktop computer for the testing? What screen size did you use? Please provide these details.
Line 128: re: “implicating the PFC as an important brain structure involved in cognitively-based inhibitory function”. While this is correct – and clarifies that you are not measuring the overall executive control with the GNG task, contrary to what you have indicated in the introduction – I am still not sure about the reasons that lead you to focus on the PFC. Please explain.
Lines 132-134: the authors indicated that “physical activity and dietary behavior were recorded before the experiment using a Chinese nutrition action program (CNAP) questionnaire.” Do you have a reference for this questionnaire? Is it a valid and reliable measure of physical activity and dietary behaviour in adult women?
Line 137: re: “These assessments occurred over the experimental testing period.” not very clear when these assessments occurred? Did they take place done before the first testing session; before and between the first and the second testing sessions; or before, between and after the first and second testing sessions?
Lines 138-139: “Nutrition records were analyzed using the Sports Nutrition Centre of the National Research Institute of Sports Medicine (NRISM) dietary and nutritional analysis system (version 3.1).” What type of analyses did you do? This is not specified in the Statistical Analysis section.
Lines 144-145: re: “The degrees of freedom were corrected using the Huynh FeldtEpsilon when the assumption of sphericity was violated”. Whilst this is a sensitive approach to avoid the increasing risk of a Type I error, please note that the F-statistic and the degrees of freedom have never been reported in the results.
Results
Line 152: I am still not sure of whether daily activity was measured for three days across each condition or three days in total. Please clarify.
Line 153-154: how did you get to the composition of nutrients and total calories?
Line 158: In the Statistical Analysis section, you wrote that ‘condition of oxygen’ identified NOR or HYP, and that ‘exercise’ identified rest or exercise.What is the variable ‘time’?
Line 168-169: from what you wrote – “There were significant time main effects on RT (p = 0.011, η2p = 0.203) and condition (p = 0.043, η2p = 0.134)” – it seems that time had a main effect on condition. Does it mean that a different time point significantly modified the condition?! This would not make very much sense. Maybe the condition had a significant main effect on RT? Please clarify.
Lines 170-171: the authors stated that “these results indicated that both moderate exercise and short-term hypoxia improved reaction time when compared with rest and normoxia.” Unfortunately, this does not come through very clearly from what they reported so far.
Lines 173-175: I am not sure why some correlation values are presented. The authors did not mention they were running correlations in the Statistical Analysis section.
Discussion
Line 183: “these alterations not being associated with changes in SpO2.” Since no background was provided on SpO2, it is not clear why this would be relevant.
Lines 186-189: From the sentence it is not clear whether the authors compared two studies on the same outcome. They stated that cognition in their study no different at rest under the two condition, and that this would contradict the results from another study where pulse oxymetry saturation was found different by study condition at baseline. I am a bit confused.
Lines 195-197: I am not sure why “…have higher average of SpO2 than males [32]. Males and females may have different physiological changes in response to severe hypoxia, which are still controversial and need to be clarified in future investigations” is under the section “cognitive functions at rest under hypoxia”. Please clarify.
Line 200: which benchmark did you use to determine that η2p = X represented a large acute effect of exercise on cognition in this sample of women? Please explain and provide relevant reference.
Line 200-201:“It has long been acknowledged that estrogen has beneficial effects on CBF.” This sentence seems to need a reference.
Lines 215-217: the authors stated: “On the contrary, Komiyama et al. [5] evaluated Go/NoGo task performance during normoxic and hypoxic exercise with a relative intensity at 50% VO2peak, which produced a lesser physical response of HRmax (~60%) than our study.” I am not sure why this is supposed to be the contrary of “SpO2was reduced to 80 ± 6% during exercise under hypoxia.”
Line 221: ”…“no attenuation” after exercise in the present study”. Wha did not show ‘attenuation’ in your study?
Lines 221-222: “…supported by, at least partially, the fact that women have fundamentally higher basal CBF.” I believe you have no evidence in support of this statement.
Line 228: re: “uncertain due to select methodological issues”. Please clarify what you mean.
Line 230: re: “The task difficulty and testing time might be too low to cause”. It is difficult for the reader to understand whether or not this could be the case, because the authors did not provide any specific details regarding the cognitive test that might indicate the overall task difficulty. It would be very beneficial for the reader, and valuable for future research, if you included more details regarding the GNG used in your study. Perhaps the use of a diagram would be beneficial.
Line 233: “Our results suggest that the hypoxic condition (12% O2) was not sufficient to affect cognitive performance.” Indeed, there was no change in the accuracy rate, but response time seemed to improve over time. Did the hypoxic condition always followed the normoxic condition in your study? Do you think that an improvement in the cognitive performance could possibly be due to a learning effect?
Conclusion
The conclusion section is minimal. Perhaps you can expand this by including the future direction for research located in different paragraphs of your Discussion section.
Minor
Introduction
Line 43: the authors stated that hypoxia “has notable cognitive-related consequences”.I suggest the authors specify here that the consequences of hypoxia on cognition appear to be negative, according to what is written in the following sentence.
Line 50: perhaps you might want to include reference 14 after the phrase “Based on the benefits caused by hypoxic exercise on physical health”
Lines 56-57: re: “Considering the neuroprotective role of estrogen in response to hypoxia”. Could you please provide more details on this aspect?
Line 64: the authors refer to “moderate exercise benefits and hypoxia-related cognitive risks” as well-established, but the evidence they provided in support of this is quite thin.
Line 68: the acronym MICE has not been introduced and has not been used consistently throughout. I suggest removing it.
Methods
Line 75: “Volunteers were publicly recruited via local media.” Where were they recruited from?
Line 100: “After a session of preliminary testing and familiarization with the executive cognition task”. What did the familiarisation consisted in?
Line 103: what is PIO2?
Line 103 and 187: FIO2has been spelled differently in the previous section (i.e., FiO2).
Line 107: Grammar issue re: “either one of the experimental conditions in which they were blinded to”.
Line 108: what is PPO?
Line 125: “Konishi and colleagues (2001)” seems to be a different referencing style from the one required by ijerph.
Line 136: what do you mean by "tracking days"? it is probably sufficient to write “days”.
Line 142: acronyms for normoxia (NOR) and hypoxia (HYP) are identified, but inconsistently used troughout the paper.
Line 146: the authors wrote that they used partial eta-squared to determine the effect size. However, the notation reported was eta-squared (η²) rather partial eta-squared (ηp² or η²p).
Line 149: is there any particular reason why the software used for statistical analysis comes with an hyperlink to https://ibm-spss-statistics-base.en.uptodown.com/windows?
Results
Line 157-159: grammar issue – I suggest simplifying the sentence by putting the subject first, then the verb, and the object last. E.g., “Condition (stats) and time (stats) had a significant main effect on HR change.” Please note that the results section presents similar issues throughout. I suggest a thorough revision of this section.
Line 163: there are some issues in the formatting of Table 1. Some values seem to be bolded, but there is no note which might explain why. Also, there are some missing spaces between numbers and signs; in other cases, it seems there are double spaces. Please amend as appropriate.
Line 166: Grammar issue regarding “less than 0.01 under rest vs. exercise during normoxia and hypoxia”. Please correct.
Line 167: Grammar issue – Figure 2 shows the results…
Line 168: Perhaps rephrase with “time had a significant main effect on RT…”, although I am still not sure what ‘time’ represents in your study.
Line 169: “No interaction effect was found between exercise and condition” …on reaction time?
Line 171: “the accurate rate”; do you mean the accuracy rate?
Line 172: grammar issue – use of neither.
Line 176: 1A should be in brackets.
Discussion
Line 179: grammar issue re: “was to, among young sedentary women, examine” – please avoid breaking up the verb with parenthetical commas. This would be more readable if it was written as “the purpose of the study was to examine the acute effects of …, among young sedentary women.”
Lines 181-182: I suggest deleting “, for young sedentary females,”
Line 190: please move “when compared to normoxia” after "in young sedentary women".
Line 191: Grammar issue regarding “effect mostly concentrated on young…”
Line 195: Grammar issue regarding “Relatively, in young adults, females have…”
Line 201: what is PaO2?
Lines 210 and 241: Grammar issue – please delete “of” after “Despite” as this is grammatically incorrect.
Line 215: “SpO2was reduced to 80 ± 6% during exercise under hypoxia” …compared to what?
Line 218: “During exercise under hypoxia, SpO2 was reduced to 77 ± 3%.” In what study? And compared to what?
Lines 218-220: “The magnitude of change in the Go/NoGo response to exercise under the similar severe hypoxia was enhanced in young women in the present study…” It is not clear what this is compared to.
Line 223-225:“Yet, to address the discrepant findings and account for differences in cognitive function, both absolute and relative exercise intensities should be considered in future research.” This seems to belong to the conclusion/future direction section.
Author Response
Dear editor,
Thanks for giving us the opportunity to revise this manuscript. As suggested, we have gone through all questions and revised point by point.
Reviewer comments for “Severe Hypoxia Does Not Offset the Benefits of Exercise On Cognitive Function in Sedentary Young Women”
Reviewer 1
General comments
The authors investigated the acute effect of acute moderate-intensity continuous exercise performed under hypoxia on cognition, in young women. Considering that there is a gap in research focusing on these aspects, the study is timely. Another positive aspect of this study is the relatively large sample size for this type of investigation (n=30 all female). Previous studies included up to 30 participants, but the sample sex was often mixed or included just male subjects.
Shortcomings of this study include: i) the lack of a comprehensive rationale for research; ii) some gaps regarding the methods, especially regarding the description of the cognitive measure and the statistical analysis; iii) the lack of clarity in presenting some of the results. Partially due to the highlighted shortcomings, the discussion section appears somewhat limited. Please refer to the specific comments for further details.
Response: We are very appreciative for your comments. We list each of your comments, and provide our responses to each of your comments (in blue font).
Specific comments
Major
Introduction
Q1. Line 45: The authors should expand background on the relationship between hypoxia and cognitive functions. The current version is quite light and does not allow the reader to understand how previous studies (references 5, 15, 16 and 17) have explored the topic; what study design have they adopted; what sample age and size; what measures of cognition have been used. Additionally, some relevant articles have been overlooked (see, McMorris, T., Hale, B. J., Barwood, M., Costello, J., & Corbett, J. (2017). Effect of acute hypoxia on cognition: A systematic review and meta-regression analysis. Neuroscience & Biobehavioral Reviews, 74, 225-232 for review).
Response: Thank you for your reminder. The references were added and the paragraph has been rewritten as follows:
More recently, an emerging line of research has shown that, apart from the benefits of improving cardiorespiratory fitness in sedentary populations, hypoxic exercise seems to be a novel treatment strategy for weight loss and comorbidities in obese subjects [16-20]. However, hypoxia itself, may induce negative cognitive-related consequences as severity increases [4, 5]. Insufficient oxygen is delivered to the brain during exercise under hypoxia, therefore leading to inadequate cerebral oxygenation and cerebral blood flow (CBF) [21]. As pulse oximetry saturation (SpO2) and cerebral oxygenation react similarly in response to hypoxia, SpO2 is generally used to indicate hypoxia degree and can also be utilized as an important predictor for cognitive function under hypoxia [22]. During moderate exercise under severe hypoxia, SpO2 decrement has been reported to attenuate exercise-induced cognitive improvement and increase reaction time in a Go/NoGo task in males [5], suggesting that hypoxia, especially severe hypoxia, may offset the benefits of exercise on cognition. Nevertheless, the findings regarding the cognitive responses to moderate exercise under severe hypoxia are inconsistent in the existing literature [5, 23, 24]. Using similar Go/NoGo Tasks (GNG) and the same fraction of inspired oxygen (FIO2 = 0.12, simulating an altitude equivalent to 4000 m), either unaffected [25] or improved [4, 5, 24, 26] cognitive performance in response to exercise has been reported. The discrepant results may be caused by differences in exercise intensities, subjects’ demographic characteristics (e.g., sex, physical fitness level and health condition), the timing for cognitive task administration and task difficulty [4, 5, 23, 24, 27, 28]. Based on the benefits caused by hypoxic exercise on physical health [1, 14, 17, 20], it is indispensable to clarify whether exercise under severe hypoxia has any adverse effects on cognitive function.
Q2. Line 49: the authors refer to previous study exploring hypoxia and cognition to have gotten to inconsistent findings, but the reader is left with no details on what these findings were. What did they found? Why is this inconsistent?
Response: As suggested, this paragraph has been rewritten as mentioned above.
Q3. Lines 52-53: “Complex cognitive processes, such as executive control, error processing, and inhibitory control, are partially regulated by the prefrontal cortex (PFC)”. Please define executive control, error processing and inhibition control with references. Also please explain why it is important to specify that these are regulated by the prefrontal cortex – considering that your study is not trying to measure anything related to a specific region of the brain, it might be sufficient to specify what the cognitive processes of interest are.
Response: Agree. To be more specific, this paragraph was rewritten as “Complex cognitive processes, such as executive control, error processing, and inhibitory control, are partially regulated by the prefrontal cortex (PFC) [29, 30], which is likely to be activated when participants performed the Go/NoGo task [25, 26, 31-34]. Previous research aimed at investigating the neural underpinnings of Go and NoGo performance demonstrated, via functional magnetic resonance imaging (fMRI), that PFC was associated with the inhibitory component (NoGo) of this task, implicating that the GNG task is involved in cognitively-based inhibitory function [35]”. Four references were added.
26. Komiyama, T., Sudo, M., Higaki, Y., Kiyonaga, A., Tanaka, H., & Ando, S., Does moderate hypoxia alter working memory and executive function during prolonged exercise? Physiology & behavior, 2015(139): p. 290-296.
25. Sudo, M., et al., Executive function after exhaustive exercise. Eur J Appl Physiol, 2017. 117(10): p. 2029-2038.
32. Harada, T., Okagawa, S., & Kubota, K., Jogging improved performance of a behavioral branching task: implications for prefrontal activation. Neuroscience research, 2004. 49(3): p. 325-337.
31. Wager, T.D., Sylvester, C. Y. C., Lacey, S. C., Nee, D. E., Franklin, M., & Jonides, J., Common and unique components of response inhibition revealed by fMRI. Neuroimage, 2005. 27(2): p. 323-340.
Q4. Lines 53-54: “The Go/NoGo task provides a simple computerized test to assess these PFC-dependent cognitive processes”. This sentence seems to be better suited in the Methods section. Also, I think the authors should be more specific on what the GNG actually measures: previous studies highlighted that it mainly taps response inhibition and attention, but I am not sure the go/no-go task is an accurate measure for the overall executive control.
Response: As suggested, this sentence has been moved to the Methods section. In addition, according to Konishi et al.’s study (1998), the Go/NoGo task is mainly involved in cognitively-based inhibitory function.
35. Konishi, S., Nakajima, K., Uchida, I., Sekihara, K., & Miyashita, Y. No‐go dominant brain activity in human inferior prefrontal cortex revealed by functional magnetic resonance imaging. European Journal of Neuroscience, 1998. 10(3): p. 1209-1213.
Q5. Line 61-63: In what way the “phase-specific differences in gonadal steroids has been shown to influence various cognitive processes, such as visual and spatial processing, and executive functions”? Please specify.
Response: As suggested, the statement was rephrased as “For example, increased emotional memory has been observed in the phase of menstrual cycle with higher progesterone level [39]. Moreover, follicle-stimulating hormone has been found to be negatively, whereas luteinizing hormone positively, correlated to visuospatial ability [47], and positive correlations between estradiol levels and paired-association learning have also been documented [48].”
Methods
Q6. Line 101: please specify how the randomization was performed.
Response: The randomization was performed using SPSS software through the function of “Random sample of cases”. Further, we added the description of our study design in the section of Methods as “We adopted a single-blinded crossover design, with the subjects being unaware of the oxygen condition, and the two experimental trials were performed in a counterbalance order to offset any potential learning effects.”
Q7. Lines 110-111: Please provide more details on the protocol of the cognitive task. Particularly please explain how participants completed the cognitive task during exercise. What was the setup? (also see major comment on Lines 122-124)
Response: The details about the protocol of cognitive task were added in “The Go/NoGo task” section as follows: “In this study, cognitive performance was evaluated through the Go/NoGo task, which provides a simple computerized test to assess these PFC-dependent cognitive processes. The reaction time and accuracy rate were collected and analyzed using an E-prime program which was installed on a 15’6 inch Lenovo B560 laptop. The laptop was placed on a portable desk in front of the cycle ergometer that can be easily moved to subjects for the cognitive task during exercise. One cognitive task included 40 trials and took approximately 2 minutes to complete. For each stimulus, two symbols would take turn to show on the middle of the screen (i.e., a square printed in red or blue color, and then followed by a number or letter printed in black color). A combination of a red square followed by a number or a combination of a blue square followed by a letter indicated the “Go” signal, in which subjects had to respond by pressing the “F” button on the keyboard with the left index finger. If the stimulus appeared in a contrary combination of a red square followed by a letter or blue square followed by a number, “NoGo” signal was presented. Subjects needed to respond by pressing the “J” button with the right index finger. The stimulus disappeared from the screen when a response was given, or else remained on screen for 2 s. A fixation cross would appear on screen for 2 s after each trial as an inter-stimulus interval. The chances of responding “Go” and “NoGo” are fifty percent, respectively. In this study, the average response time (ms) and the accuracy rate (%) form all trials were calculated to measure inhibitory control.”
Q8. Line 112: why did you measure pulse oximetry saturation (SpO2)? There is no background on why you have included this measure.
Response: As suggested, the background information regarding to SpO2 was added in the Introduction: “As pulse oximetry saturation (SpO2) and cerebral oxygenation reacted similarly in response to hypoxia, SpO2 is generally used to indicate hypoxia gradation and can also be utilized as an important predictor for cognitive function under hypoxia [22].”
22. McMorris, T., et al., Effect of acute hypoxia on cognition: A systematic review and meta-regression analysis. Neurosci Biobehav Rev, 2017. 74(Pt A): p. 225-232.
Q9. Lines 119-120: Figure 1 needs some work: i) from the figure it is not clear how to locate Normoxia; ii) do the arrows represent the start and end of the GNG task? I suggest replacing them with a gray pattern. The arrows make it seem you have measured participants twice at rest and twice during exercise, which I suppose you did not do; iii) Please indicate "time (min)" under the X axis. Also, please note that the numbers under the figure do not follow a logical order. Indeed, they appear to be placed at random: the space between 10 and 12 is greater than the one between 0 and 10, and the one between 15 and 22; iv) the acronym PPO should be defined in the notes/legend.
Response: Thank you very much for the correction. Figure 1 has been revised as suggested.
Q10. Lines 122-124: the description of the cognitive task used to measure the main outcome of the study very limited and currently lacks important information: did you use the original paradigm developed by Donders in 1969 or is it a modified version? If it is a modified version what program did you use to modify the task? What parameters did you use? How many trials? How many blocks? How much time did each trial last? How much time between each trial? Did you have a fixation cross on the screen between trials? Did the participant have a practice block? How long did the practice go for? Did you use the ‘raw’ data for your analysis (i.e., response time to each individual trial) or made an average of them? Did you use the response time from go tasks or both go and no-go trials? How did you treat premature responses? Did you extract the accuracy rate from all trials or only the no-go trials? Did you use a laptop or a desktop computer for the testing? What screen size did you use? Please provide these details.
Response: The description of the Go/NoGo task was added as previously mentioned. Thank you very much.
Q11. Line 128: re: “implicating the PFC as an important brain structure involved in cognitively-based inhibitory function”. While this is correct – and clarifies that you are not measuring the overall executive control with the GNG task, contrary to what you have indicated in the introduction – I am still not sure about the reasons that lead you to focus on the PFC. Please explain.
Response: Sorry that we did not make it clear in our writing. This description was moved to the Introduction section as background information. Indeed, we were not trying to measure anything related to a specific region of the brain. However, according to the literature, “Previous research aimed at investigating the neural underpinnings of Go and NoGo performance demonstrated, via functional magnetic resonance imaging (fMRI), that PFC was associated with the inhibitory component (NoGo) of this task, implicating that the GNG task is involved in cognitively-based inhibitory function [35]”, PFC is likely to be activated during the GNG task and we mentioned PFC as a background information.
35. Konishi, S., Nakajima, K., Uchida, I., Sekihara, K., & Miyashita, Y. No-go dominant brain activity in human inferior prefrontal cortex revealed by functional magnetic resonance imaging. European Journal of Neuroscience, 1998. 10(3): 1209-1213.
Q12. Lines 132-134: the authors indicated that “physical activity and dietary behavior were recorded before the experiment using a Chinese nutrition action program (CNAP) questionnaire.” Do you have a reference for this questionnaire? Is it a valid and reliable measure of physical activity and dietary behaviour in adult women?
Response: Thank you for your reminder. A pedometer (Ymax SW-200 digiwalker, Japan) was used to assess daily physical activity, which has previously been validated [1, 58]. We are sorry that the validity and reliability of this tool cannot been found in the resources in both Chinese and English literature, however, the CNAP questionnaire has been extensively used in previous studies [1, 59].
1. Kong, Z., et al., High-Intensity Interval Training in Normobaric Hypoxia Improves Cardiorespiratory Fitness in Overweight Chinese Young Women. Front Physiol, 2017. 8: p. 175.
58. Duncan, S.J., Schofield, G., Duncan, E. K., & Hinckson, E. A., Effects of age, walking speed, and body composition on pedometer accuracy in children. Research Quarterly for Exercise and Sport, 2007. 78(5): p. 420-428.
59. Kong, Z.; Sun, S.; Liu, M.; Shi, Q. Short-Term High-Intensity Interval Training on Body Composition and Blood Glucose in Overweight and Obese Young Women. J. Diabetes Res. 2016, 2, 1–9.
Q13. Line 137: re: “These assessments occurred over the experimental testing period.” not very clear when these assessments occurred? Did they take place done before the first testing session; before and between the first and the second testing sessions; or before, between and after the first and second testing sessions?
Response: Sorry for the unclear description. It should be “Physical activity and dietary behavior were recorded for 2 days (one day before and the day of the experiment) during each experimental trial.”
Q14. Lines 138-139: “Nutrition records were analyzed using the Sports Nutrition Centre of the National Research Institute of Sports Medicine (NRISM) dietary and nutritional analysis system (version 3.1).” What type of analyses did you do? This is not specified in the Statistical Analysis section.
Response: “Paired t-tests were conducted to confirm the consistency of physical activity and dietary behaviour.” This sentence has been added in the Statistical Analysis section.
Q15. Lines 144-145: re: “The degrees of freedom were corrected using the Huynh FeldtEpsilon when the assumption of sphericity was violated”. Whilst this is a sensitive approach to avoid the increasing risk of a Type I error, please note that the F-statistic and the degrees of freedom have never been reported in the results.
Response: Thank you for your reminder. The F-statistic and the degrees of freedom have been reported in the revised manuscript.
Results
Q16. Line 152: I am still not sure of whether daily activity was measured for three days across each condition or three days in total. Please clarify.
Response: It should be “Physical activity and dietary behavior were recorded for 2 days (one day before and the day of the experiment) during each experimental trial.”
Q17. Line 153-154: how did you get to the composition of nutrients and total calories?
Response: The data (i.e. the weight and composition of food and beverage consumed) obtained from the participants with (CNAP) questionnaire were inputted into the Sports Nutrition Centre of the National Research Institute of Sports Medicine (NRISM) dietary and nutritional analysis system (version 3.1), then the computer software would calculate the macronutrient proportion and total calorie intake results automatically.
Q18. Line 158: In the Statistical Analysis section, you wrote that ‘condition of oxygen’ identified NOR or HYP, and that ‘exercise’ identified rest or exercise. What is the variable ‘time’?
Response: Sorry for the confusion. The variable ‘time’ referred to rest and exercise, thus ‘time’ means ‘exercise’ here. To avoid vagueness in this statement, we used ‘exercise’ to replace ‘time’ throughout the paper in the revised version.
Q19. Line 168-169: from what you wrote – “There were significant time main effects on RT (p = 0.011, η2p = 0.203) and condition (p = 0.043, η2p = 0.134)” – it seems that time had a main effect on condition. Does it mean that a different time point significantly modified the condition?! This would not make very much sense. Maybe the condition had a significant main effect on RT? Please clarify.
Response: Thank you for your reminder. It should be: “There were significant main effects of exercise [F(1,29) = 8.336, p = 0.011, η2p = 0.203] and condition [F(1,29) = 5.425, p = 0.043, η2p = 0.134] on RT.”
Q20. Lines 170-171: the authors stated that “these results indicated that both moderate exercise and short-term hypoxia improved reaction time when compared with rest and normoxia.” Unfortunately, this does not come through very clearly from what they reported so far.
Response: As suggested, more information was provided in the section of Results and the statement has been rephrased: “There were significant main effects of exercise [F(1,29) = 8.336, p = 0.011, η2p = 0.203] and condition [F(1,29) = 5.425, p = 0.043, η2p = 0.134] on RT, whereas no interaction effect was found between exercise and condition [F(1,29) = 0.524, p = 0.573, η2p = 0.011] on RT. Further analyses demonstrated that resting RT under hypoxia was faster than under normoxia (p = 0.012), while RT during exercise was faster than at rest in normoxia (p = 0.027). These results indicated that both moderate exercise and short-term hypoxia improved reaction time.”
Q21. Lines 173-175: I am not sure why some correlation values are presented. The authors did not mention they were running correlations in the Statistical Analysis section.
Response: Sorry that we forgot to include such information in the Statistical Analysis section. The statement was rephrased “In addition, Pearson’s product-moment correlation coefficients were calculated to evaluate the correlations between the changes of RT and SpO2.”
Discussion
Q22. Line 183: “these alterations not being associated with changes in SpO2.” Since no background was provided on SpO2, it is not clear why this would be relevant.
Response: The background information regarding to SpO2 has been added in the Introduction as mentioned previously.
Q23. Lines 186-189: From the sentence it is not clear whether the authors compared two studies on the same outcome. They stated that cognition in their study no different at rest under the two condition, and that this would contradict the results from another study where pulse oxymetry saturation was found different by study condition at baseline. I am a bit confused.
Response: Thank you for pointing this out. The discussion of the 4.1 section was rewritten throughout.
“In the present study, at the beginning of each experimental trial, we measured cognitive function at rest under normoxia and hypoxia. In contrast to our hypothesis, reaction time was decreased with an unaffected accuracy rate after resting under hypoxia for 10 min, indicating that the response speed was improved after hypoxic exposure. This result is consistent with some previous findings, suggesting that hypoxia did not impair cognitive performance [4, 5]. However, using similar hypoxic exposure levels (FIO2 = 0.12 or 0.125), there are several studies reporting unchanged [28] or slower reaction time [23], and decreased accuracy rate [24] in male subjects [24] and mixed subjects [23, 28]. Compared to these studies [23, 28], the female subjects in the present study showed a relatively higher SpO2 value. The different SpO2 response to the same hypoxia level between males and females may, in part, be explained by the stronger resistance to hypoxia in women [46]. This may have occurred given the fact that we conducted this study during the early follicular stage of the female subjects when estrogen hormones were at a relatively high level, and thus, may provide a neuroprotective response to hypoxia [45]. In support of this notion, various cognitive parameters are heightened during the follicular stage, at times of higher estrogen or progesterone, were reported in previous studies [39, 48, 61].”
Q24. Lines 195-197: I am not sure why “…have higher average of SpO2 than males [32]. Males and females may have different physiological changes in response to severe hypoxia, which are still controversial and need to be clarified in future investigations” is under the section “cognitive functions at rest under hypoxia”. Please clarify.
Response: Thank you for your reminder. We have revised the sentence and moved it to the Conclusion.
Q25. Line 200: which benchmark did you use to determine that η2p = X represented a large acute effect of exercise on cognition in this sample of women? Please explain and provide relevant reference.
Response: The statements was rephrased “The effect size was considered small if η2p < 0.06 and large if η2p > 0.14 [60].” A reference was added after the statement.
Cohen, J., Statistical power analysis for the social sciences. 1988.
Q26. Line 200-201:“It has long been acknowledged that estrogen has beneficial effects on CBF.” This sentence seems to need a reference.
Response: Given that the Introduction and discussion sections were rewritten, this sentence was moved to the Introduction and one reference was added.
Sundström Poromaa, I. and M. Gingnell, Menstrual cycle influence on cognitive function and emotion processing-from a reproductive perspective. Front Neurosci, 2014. 8: p. 380.
Q27. Lines 215-217: the authors stated: “On the contrary, Komiyama et al. [5] evaluated Go/NoGo task performance during normoxic and hypoxic exercise with a relative intensity at 50% VO2peak, which produced a lesser physical response of HRmax (~60%) than our study.” I am not sure why this is supposed to be the contrary of “SpO2was reduced to 80 ± 6% during exercise under hypoxia.”
Response: Sorry to use an unclear expression. We have revised the paragraph as follows:
“Under normoxia and hypoxia, the present study used the same absolute intensity (45% PPO), which was equivalent to 73% HRmax under normoxia and 80% HRmax under hypoxia. Differently, Komiyama et al. [5] evaluated Go/NoGo task performance during nomoxic and hypoxic exercise with a relative intensity at 50% VO2peak (corresponding to ~60% of HRmax), which was lower than our study. Despite the greater exercise intensity, the reduction of SpO2 was not enlarged in the female subjects of our study compared to the male subjects in their study[5], both decreasing from 87% to 77%.”
Q28. Line 221: ”…“no attenuation” after exercise in the present study”. What did not show ‘attenuation’ in your study?
Response: The statement was unclear, thus it was rewritten as “Furthermore, the observation of that severe hypoxia did not attenuate exercise-induced benefits in cognition in the present study is ...”.
Q29. Lines 221-222: “…supported by, at least partially, the fact that women have fundamentally higher basal CBF.” I believe you have no evidence in support of this statement.
Response: As a matter of fact, a few researchers have studied this topic. Women have higher basal CBF than men under both normoxia and hypoxia have been observed. We have added the following sentences in our revised manuscript: “Furthermore, the observation of which severe hypoxia did not attenuate exercise-induced benefit in cognition in young women in the present study is supported by, at least partially, the aforementioned fact that women have fundamentally higher basal CBF than men under both normoxic [63, 64] and hypoxic conditions [65] because of the increased cerebral artery vasodilation stimulated by the increase of estrogen [42-44, 65]”. And several references have been added.
63. Rodriguez, G., Warkentin, S., Risberg, J., & Rosadini, G. (1988). . , 8(6), 783-789., Sex differences in regional cerebral blood flow. Journal of Cerebral Blood Flow & Metabolism, 1988. 8(6): p. 783-789.
42. Ospina, J.A., Krause, D. N., & Duckles, S. P., 17β-Estradiol increases rat cerebrovascular prostacyclin synthesis by elevating cyclooxygenase-1 and prostacyclin synthase. Stroke, 2002. 33(2): p. 600-605.
43. Ospina, J.A., Duckles, S. P., & Krause, D. N., 17β-Estradiol decreases vascular tone in cerebral arteries by shifting COX-dependent vasoconstriction to vasodilation. American Journal of Physiology-Heart and Circulatory Physiology., 2003.
44. Sobrino, A., Oviedo, P. J., Novella, S., Laguna-Fernandez, A., Bueno, C., Garcia-Perez, M. A., ... & Hermenegildo, C., Estradiol selectively stimulates endothelial prostacyclin production through estrogen receptor-a. J Mol Endocrinol, 2010. 44(4): p. 237-246.
64. Esposito, G., Van Horn, J. D., Weinberger, D. R., & Berman, K. F., Gender differences in cerebral blood flow as a function of cognitive state with PET. Journal of Nuclear Medicine, 1996. 37(4): p. 559-564.
65. Peltonen, G.L., Harrell, J. W., Rousseau, C. L., Ernst, B. S., Marino, M. L., Crain, M. K., & Schrage, W. G., Cerebrovascular regulation in men and women: stimulus‐specific role of cyclooxygenase. Physiological reports, 2015. 3(7).
Q30. Line 228: re: “uncertain due to select methodological issues”. Please clarify what you mean.
Response: To be more specific, the sentence was revised as: “… still remains uncertain due to diverse methodological issues (e.g., differences in cognitive task-specific measure, subjects, severity of hypoxia, hypoxic exposure time).”
Q31. Line 230: re: “The task difficulty and testing time might be too low to cause”. It is difficult for the reader to understand whether or not this could be the case, because the authors did not provide any specific details regarding the cognitive test that might indicate the overall task difficulty. It would be very beneficial for the reader, and valuable for future research, if you included more details regarding the GNG used in your study. Perhaps the use of a diagram would be beneficial.
Response: The details regarding the GNG task used in this study were added in the Methods section as mentioned before. Thank you very much.
Q32. Line 233: “Our results suggest that the hypoxic condition (12% O2) was not sufficient to affect cognitive performance.” Indeed, there was no change in the accuracy rate, but response time seemed to improve over time. Did the hypoxic condition always followed the normoxic condition in your study? Do you think that an improvement in the cognitive performance could possibly be due to a learning effect?
Response: Firstly, all subjects practiced the cognition task multiple times until the number of errors trials in each practice session were less than 5 (one practice session includes 40 trials), suggesting that they were familiar with the cognitive task before participating in the formal experimental trial. Secondly, we adopted a single-blinded crossover design and the two experimental trials were performed in a counterbalance order, which means half of the subjects performed hypoxic trial first whereas the other half conducted the normoxic trial first so that the learning effects were counteracted. Based on these two reasons, the cognition improvement yielded in the present study was unlikely to be caused by a learning effect. This information has been added to the revised manuscript.
Q33 Conclusion
The conclusion section is minimal. Perhaps you can expand this by including the future direction for research located in different paragraphs of your Discussion section.
Response: Thank you for your suggestion. Several sentences about future direction were added in the Conclusion: “In addressing the discrepant findings and accounting for differences in cognitive function, both absolute and relative exercise intensities should be considered in future research. As males and females seem to have different physiological and cognitive responses to severe hypoxia, whether sex-specific hormones influence this process need to be further clarified. Moreover, future studies may also consider adopting different exercise modes (e.g., high-intensity interval exercise, resistance exercise) and using more sophisticated and higher-order cognitive tasks to evaluate cognitive function, meanwhile, monitoring CBF and cerebral oxygenation profiles to reveal the potential mechanisms”.
Minor
Introduction
Q34. Line 43: the authors stated that hypoxia “has notable cognitive-related consequences”. I suggest the authors specify here that the consequences of hypoxia on cognition appear to be negative, according to what is written in the following sentence.
Response: As suggested, the sentence was revised as: “However, hypoxia itself, may has negative cognitive-related consequences …”.
Q35. Line 50: perhaps you might want to include reference 14 after the phrase “Based on the benefits caused by hypoxic exercise on physical health”
Response: Thank you, several references were added.
Q36. Lines 56-57: re: “Considering the neuroprotective role of estrogen in response to hypoxia”. Could you please provide more details on this aspect?
Response: As suggested, more information about relationships between sex-specific hormones and cognition has been added at the end of this paragraph: “For example, increased emotional memory has been observed in the phase of menstrual cycle with higher progesterone level [39]. Moreover, follicle-stimulating hormone has been found to be negatively, whereas luteinizing hormone positively correlated to visuospatial ability [47], and positive correlations between estradiol levels and paired-association learning have also been documented [48].”
Q37. Line 64: the authors refer to “moderate exercise benefits and hypoxia-related cognitive risks” as well-established, but the evidence they provided in support of this is quite thin.
Response: Thank you for your reminder. The relevant references used in this studies were added after the statement.
Q38. Line 68: the acronym MICE has not been introduced and has not been used consistently throughout. I suggest removing it.
Response: Agree, the acronym MICE has been removed throughout the manuscript.
Methods
Q39. Line 75: “Volunteers were publicly recruited via local media.” Where were they recruited from?
Response: Volunteers were publicly recruited via flyers posted on the campus of the University of Macau. This has been added to our revised methods section.
Q40. Line 100: “After a session of preliminary testing and familiarization with the executive cognition task”. What did the familiarisation consisted in?
Response: The details about the familiarization were added in the Methods: “During the familiarization, subjects learned the experimental procedures and practiced the cognition task multiple times until the number of errors trials in each session were less than 5 (one practice session includes 40 trials)”.
Q41. Line 103: what is PIO2?
Response: PIO2 is inspiratory oxygen pressure, and the notation was added.
Q42. Line 103 and 187: FIO2has been spelled differently in the previous section (i.e., FiO2).
Response: Thank you for your reminder. The spelling of FIO2 has been checked and unified throughout the manuscript.
Q43. Line 107: Grammar issue re: “either one of the experimental conditions in which they were blinded to”.
Response: The sentence was revised as “… participants rested quietly in a seated position for 10 minutes with a mask connected to either normoxia or hypoxia condition in which they were blinded to”.
Q44. Line 108: what is PPO?
Response: PPO is peak power output, and the notation was added.
Q45. Line 125: “Konishi and colleagues (2001)” seems to be a different referencing style from the one required by ijerph.
Response: Thank you, we have fixed this issue.
Q46. Line 136: what do you mean by "tracking days"? it is probably sufficient to write “days”.
Response: This paragraph has been rewritten, and the relevant sentence were revised as “Physical activity and dietary behavior were recorded for 2 days (one day before and the day of the experiment) during each experimental trial”.
Q47. Line 142: acronyms for normoxia (NOR) and hypoxia (HYP) are identified, but inconsistently used throughout the paper.
Response: Thank you for pointing this out. The acronyms of normoxia (NOR) and hypoxia (HYP) are adopted to represent the two experimental trails of the present study, but was inconsistently used in some places. We have checked the use of NOR and HYP in the manuscript carefully.
Q48. Line 146: the authors wrote that they used partial eta-squared to determine the effect size. However, the notation reported was eta-squared (η²) rather partial eta-squared (ηp² or η²p).
Response: Thank you for your reminder. We have made the correction, and the notation was unified to η2p.
Q49. Line 149: is there any particular reason why the software used for statistical analysis comes with an hyperlink to https://ibm-spss-statistics-base.en.uptodown.com/windows?
Response: The hyperlink was accidently added which has no particular reason, and it has been deleted.
Results
Q50. Line 157-159: grammar issue – I suggest simplifying the sentence by putting the subject first, then the verb, and the object last. E.g., “Condition (stats) and time (stats) had a significant main effect on HR change.” Please note that the results section presents similar issues throughout. I suggest a thorough revision of this section.
Response: Thank you. We have rewritten this section as suggested.
Q51. Line 163: there are some issues in the formatting of Table 1. Some values seem to be bolded, but there is no note which might explain why. Also, there are some missing spaces between numbers and signs; in other cases, it seems there are double spaces. Please amend as appropriate.
Response: Thank you. We have corrected Table 1 as suggested.
Q52. Line 166: Grammar issue regarding “less than 0.01 under rest vs. exercise during normoxia and hypoxia”. Please correct.
Response: The notation was revised as: “…a = p value less than 0.05 under HYP vs. NOR; b = p value less than 0.01 under HYP and NOR.”
Q53. Line 167: Grammar issue – Figure 2 shows the results…
Response: Thank you for the correction. We have amended it as suggested.
Q54. Line 168: Perhaps rephrase with “time had a significant main effect on RT…”, although I am still not sure what ‘time’ represents in your study.
Response: In the present study, ‘Time effect’ actually represented ‘exercise effect’. In order to avoid confusion, “exercise” was used to replace “time” throughout the manuscript.
Q55. Line 169: “No interaction effect was found between exercise and condition” …on reaction time?
Response: Yes, we have made the correction.
Q56. Line 171: “the accurate rate”; do you mean the accuracy rate?
Response: Yes, it should be “the accuracy rate” and we have made the correction.
Q57. Line 172: grammar issue – use of neither.
Response: Agree, and done.
Q58. Line 176: 1A should be in brackets.
Response: We have made the correction as suggested.
Discussion
Q59. Line 179: grammar issue re: “was to, among young sedentary women, examine” – please avoid breaking up the verb with parenthetical commas. This would be more readable if it was written as “the purpose of the study was to examine the acute effects of …, among young sedentary women.”
Response: As suggested, we have made the correction.
Q60. Lines 181-182: I suggest deleting “, for young sedentary females,”
Response: It was deleted as suggested.
Q61. Line 190: please move “when compared to normoxia” after "in young sedentary women".
Response: The whole paragraph of 4.1 was rewritten and the sentence was deleted.
Q62. Line 191: Grammar issue regarding “effect mostly concentrated on young…”
Response: The whole paragraph of 4.1 has been rewritten.
Q63. Line 195: Grammar issue regarding “Relatively, in young adults, females have…”
Response: Thank you for your reminder. The whole paragraph of 4.1 was rewritten and this expression has been deleted.
Q64. Line 201: what is PaO2?
Response: The whole paragraph of 4.2 has been rewritten.
Q65. Lines 210 and 241: Grammar issue – please delete “of” after “Despite” as this is grammatically incorrect.
Response: It was deleted as suggested.
Q66. Line 215: “SpO2was reduced to 80 ± 6% during exercise under hypoxia” …compared to what?
Response: Thank you for your pointing this out. The whole paragraph has been amended.
Q67. Line 218: “During exercise under hypoxia, SpO2 was reduced to 77 ± 3%.” In what study? And compared to what?
Response: Thank you for your reminder. The whole paragraph has been amended.
Q68. Lines 218-220: “The magnitude of change in the Go/NoGo response to exercise under the similar severe hypoxia was enhanced in young women in the present study…” It is not clear what this is compared to.
Response: Thank you for pointing this out. This unclear sentence has been deleted and the whole paragraph of 4.2 has been rewritten.
Q69. Line 223-225:“Yet, to address the discrepant findings and account for differences in cognitive function, both absolute and relative exercise intensities should be considered in future research.” This seems to belong to the conclusion/future direction section.
Response: As suggested, this sentence has been moved to the conclusion section.
Reviewer 2 Report
Abstract:
Page 1, Line 23-25: Please ascertain the results “RT (p < 0.05, η2p = 0.234) decreased during moderate exercise when compared at rest, while a short bout of severe hypoxia improved RT (p < 0.05, η2p = 0.134).” which do not correspond to the Figure 2 (1A) illustrated on the Page 5 and the results stated on the Page 4, Lines 170-171.
Introduction:
Page 1, Line 39: Erickson et al.’ s study (2011) explored the effects of “long-term” exercise intervention on BDNF, spatial memory, and hippocampal volume in older adults. Authors should cite previous studies regarding the “acute” exercise (e.g., Ferris et al. (2007) in Med. Sci. Sports Exerc. & Tsai et al. (2014) in Psychoneuroendocrinology & Tsai et al. (2014) in Front. Behav. Neurosci.)
Page 1, Lines 40-42: Please explain why Camacho-Cardenosa et al.’s study (2018) exploring a 12 week program of exercise intervention in normoxia/ hypoxia in overweight/obese adults was cited in the present study.
Page 2, the 2nd paragraph: Two topics, Go/No-Go task and estrogen, introduced in this paragraph should be separate into two parts.
Estrogen is the main reason to support why the present study should be performed in the young women. Although this paper aims to study an interesting topic, it is necessary to have more evidence to back up the authors’ claim. For example, substantial data (e.g., estrogen levels) are required to perform the correlation analysis. Or, collecting data from young men to compare the results with the present findings are necessary.
Page 2, Line 68 & Page 3, Line 108: Please spell out the acronyms (e.g., MICE & PPO) when they are used for the first time in the body of the paper.
Methods:
Please illustrate the unit for the numbers below the line in the Figure 1.
Results:
Page 4, Lines 160-161: Based on the “Regarding SpO2, condition (p < 0.001, η2p = 0.904) and time (p < 0.001, η2p = 0.634) had significant main effects and an interaction effect (p < 0.001, η2p= 0.574)”, the post hoc test should be run and the results should be clearly described.
Since the “condition” and “exercise” are within-subject factors (Page 4, Lines 142-143), many sentences, “Regarding SpO2, condition and time had significant main effects and an interaction effect (Lines 160-161)”, “there were significant time main effects on RT and condition (Lines 168-169)”, and “neither no main effects of time and condition (Line 172)”, will confuse the readers.
Page 4, Lines 168-171: Most importantly, the results regarding the RT do not correspond to the Figure 2 (1A).
Discussion:
More evidence or experimental designs are needed to strengthen the explanation/speculations regarding the estrogen/progesterone and cerebral blood flow.
Author Response
Reviewer 2
Comments and Suggestions for Authors
Response: Thank you very much for your comments. We list each point of your comments, and provide our responses one by one (in blue font).
Abstract:
Q1. Page 1, Line 23-25: Please ascertain the results “RT (p < 0.05, η2p = 0.234) decreased during moderate exercise when compared at rest, while a short bout of severe hypoxia improved RT (p < 0.05, η2p = 0.134).” which do not correspond to the Figure 2 (1A) illustrated on the Page 5 and the results stated on the Page 4, Lines 170-171.
Response: Thank you for your pointing out this. The results in the Abstract was rephrased “RT (p < 0.05, η2p = 0.203) decreased during moderate exercise …”
Introduction:
Q2. Page 1, Line 39: Erickson et al.’ s study (2011) explored the effects of “long-term” exercise intervention on BDNF, spatial memory, and hippocampal volume in older adults. Authors should cite previous studies regarding the “acute” exercise (e.g., Ferris et al. (2007) in Med. Sci. Sports Exerc. & Tsai et al. (2014) in Psychoneuroendocrinology & Tsai et al. (2014) in Front. Behav. Neurosci.)
Response: Thank you for your suggestion. We have deleted Erickson et al.’s study (2011) and the three recommended references were added.
Q3. Page 1, Lines 40-42: Please explain why Camacho-Cardenosa et al.’s study (2018) exploring a 12 week program of exercise intervention in normoxia/ hypoxia in overweight/obese adults was cited in the present study.
Response: This paragraph illustrated the background information regarding hypoxic training and the rational of the present study. Previous studies suggested that hypoxic training can induce numerous health benefits (e.g., improving cardiovascular fitness, losing weight) and has therapeutic potential for obesity and related comorbidities, but the “hypoxic exposure” itself may impair cognitive function, that’s the reason why we are interested to further investigate whether hypoxic exercise has adverse effects on cognition. The reference was cited to show the background information.
Q4. Page 2, the 2nd paragraph: Two topics, Go/No-Go task and estrogen, introduced in this paragraph should be separate into two parts.
Response: Agree. We have separated this paragraph as suggested.
Q5. Estrogen is the main reason to support why the present study should be performed in the young women. Although this paper aims to study an interesting topic, it is necessary to have more evidence to back up the authors’ claim. For example, substantial data (e.g., estrogen levels) are required to perform the correlation analysis. Or, collecting data from young men to compare the results with the present findings are necessary.
Response: This is wonderful advice. However, we could not provide such information in the present study as we only included females and have not measured estrogen levels. We would like to extend the research scope in our future study.
Q6. Page 2, Line 68 & Page 3, Line 108: Please spell out the acronyms (e.g., MICE & PPO) when they are used for the first time in the body of the paper.
Response: Thank you for your reminder. We have added the complete spelling when we use the acronyms for the first time.
Methods:
Q7. Please illustrate the unit for the numbers below the line in the Figure 1.
Response: Thank you for your reminder. Figure 1 was revised as suggested.
Results:
Q8. Page 4, Lines 160-161: Based on the “Regarding SpO2, condition (p < 0.001, η2p = 0.904) and time (p < 0.001, η2p = 0.634) had significant main effects and an interaction effect (p < 0.001, η2p= 0.574)”, the post hoc test should be run and the results should be clearly described.
Response: Thank you for your reminder. It has been rephrased as “Regarding SpO2, condition [F(1,29) = 308.173, p < 0.001, η2p = 0.904] and exercise [F(1,29) = 52.244, p < 0.001, η2p = 0.634] had significant main effects and an interaction effect [F(1,29) = 40.930, p < 0.001, η2p= 0.574]. Further analyses indicated that, when compared to normoxia, SpO2 in hypoxia experienced significant decreases either at rest (p < 0.001) or during exercise (p < 0.001).”
Q9. Since the “condition” and “exercise” are within-subject factors (Page 4, Lines 142-143), many sentences, “Regarding SpO2, condition and time had significant main effects and an interaction effect (Lines 160-161)”, “there were significant time main effects on RT and condition (Lines 168-169)”, and “neither no main effects of time and condition (Line 172)”, will confuse the readers.
Response: Totally agree, sorry that we did not make it clear in our writing. To avoid vagueness, ‘exercise’ was used to replace ‘time’ in the revised manuscript.
Q10. Page 4, Lines 168-171: Most importantly, the results regarding the RT do not correspond to the Figure 2 (1A).
Response: Thank you for your pointing this out. We have made the correction.
Discussion:
Q11. More evidence or experimental designs are needed to strengthen the explanation/speculations regarding the estrogen/progesterone and cerebral blood flow.
Response: The discussion section was rewritten, the relevant discussion about “the estrogen/progesterone and cerebral blood flow” was strengthened. In addition, measuring sex-specific hormones and CBF was recommended as directions for future study in the Conclusion section. Thank you very much for the wonderful suggestion.
Round 2
Reviewer 1 Report
Manuscript: ijerph-438545
Title: Severe Hypoxia Does Not Offset the Benefits of Exercise On Cognitive Function in Sedentary Young Women
General comment
Dear authors, thank you for submitting a revised version of your manuscript. I think the manuscript has improved compared to the previously submitted version, although there are still some major flaws that would need attention.
Specific comments
English language editing is recommended.
Line 52: The introduction is now much clearer than before. However, please note that a reader who is not expert in measures of cognition might be wondering what a Go/NoGo task is. This was already highlighted in my previous revision. Please refer to my previous comment (Q4), but the reply from the authors is not entirely satisfactory. Perhaps try to refer to people's response inhibition ability, rather than to the test - or maybe you can do both (i.e., response inhibition reaction time (measured with Go/NoGo task))
Line 55: the acronym GNG was not introduced when Go/NoGo task was presented the first time (see line 52) and is then used inconsistently. I suggest the author make a consistent use of the acronym throughout or delete it.
Line 57: what aspect of cognitive performance? Were there any differences in the way different studies measured cognitive functions? My previous comment (Q2) has only partially been addressed.
Lines 64-68: Please note that the change made here in response to my previous comment (see Q3 and Q11) are not satisfactory. Indeed, the points arisen with my previous comments are still unanswered: 1) please define executive control, error processing and inhibition control; 2) what "is likely to be activated when the participants performed the Go/NoGo task"? The subject of this sentence is unclear as there are multiple possibilities in the precedent phrase; and 3) I am still unclear on the reasons that made the authors decide to put so much emphasis on identifying the cortical region associated with inhibitory control, if a deeper investigation of brain activity is not part of the research questions.
Lines 85-89: Importantly, none of the examples you have provided here are related to executive functions or, more specifically, response inhibition—which is what you measured. Although reference 39 provides some support to your decision to investigate cognitive functions in women, the choice of using GNG is not really supported. Indeed, Sundström Poromaa and Gingnell [39, p.8] reported: "Using tasks that test the ability to inhibit prepotent responses, Colzato et al. reported on less efficient inhibition in the late follicular phase (Colzato et al.,2010), whereas Bannbers and colleagues found no effect of men-strual cycle, either in accuracy or reaction time to a Go-NoGo task (Bannbers et al., 2012). However, using a task that probed inhibitory input control, as opposed to the Stop-signal task and Go-NoGo concerned with inhibitory output control, Colzato and colleagues demonstrated superior inhibition of return in the late follicular phase (Colzato et al., 2012)."
Line 121: How did you measure the heart rate?
Lines 155-171: The description of the task is much clearer now. This, however, highlighted that the Go/NoGo task you have used looks more like a Flanker's task than a Go/NoGo task. The Go/NoGo paradigm normally requires participants to inhibit their responses when presented with a NoGo trial (which is the reason why the trial is called NoGo). In your experiment, NoGo trials required participants to press a button. The cognitive skill that allows you to do this is a different aspect of inhibition, known as interference. Also, please note that there is no reference to the cognitive paradigm that you have used. This is clearly not based on the original paradigm developed by Donders (1969). Donders FC. On the speed of mental processes. Acta Psychologica. 1969;30:412–431. (Original work published 1868)
Results: I find the results section a bit confusing and I am afraid that other readers will also have a difficult time with it. Moreover, the collected physiological parameters are not very much integrated with the cognitive data, except from one sudden apparition of SpO2 on line 224-225 which is compared to the study conditions with correlations. I think that you need to clarify much more in the statistical analysis subsection what analysis you have conducted and why, following a logical order. Then, in the result section, the findings should be presented following the same logical order. Adding some lack of clarity, the results related to cognition are presented under the physiolocal parameters subsection.
Line 225-226: There are some graphical issues with Figure 2. The images appeared cropped in an odd way. Figure 2A presents some issues with the connectors and asterisk over NOR.
Lines 236-238: the sentence seems to say that reaction time decreased indicating that reaction time decreased.
Lines 238-239: this is somewhat conflicting with what you wrote on Lines 45-46: "However, hypoxia itself, may induce negative cognitive-related consequences as severity increases [4, 5].”
Lines 242-249: I am not sure what you are trying to say with your discussion point on SpO2. Are you trying to explain something related to your study findings regarding cognitive performance?
Lines 259-262: Please try to simplify the sentence. It sounds like you are saying the exact opposite of what you have just stated on line 259 (i.e., "be compensated by an increased CBF").
Line 275: What do you mean when you say that SpO2 was not 'enlarged'?
Author Response
General comment
Dear authors, thank you for submitting a revised version of your manuscript. I think the manuscript has improved compared to the previously submitted version, although there are still some major flaws that would need attention.
Specific comments
English language editing is recommended.
1. Line 52: The introduction is now much clearer than before. However, please note that a reader who is not expert in measures of cognition might be wondering what a Go/NoGo task is. This was already highlighted in my previous revision. Please refer to my previous comment (Q4), but the reply from the authors is not entirely satisfactory. Perhaps try to refer to people's response inhibition ability, rather than to the test - or maybe you can do both (i.e., response inhibition reaction time (measured with Go/NoGo task))
(The previous comment Q4 is “Also, I think the authors should be more specific on what the GNG actually measures: previous studies highlighted that it mainly taps response inhibition and attention, but I am not sure the go/no-go task is an accurate measure for the overall executive control.” please delete the sentences in red font after response)
Response: Thank you for pointing this out. As suggested, we have completed the revision.
2. Line 55: the acronym GNG was not introduced when Go/NoGo task was presented the first time (see line 52) and is then used inconsistently. I suggest the author make a consistent use of the acronym throughout or delete it.
Response: Thank you for pointing this out. The acronym GNG was deleted, and inconsistent use of Go/NoGo task was checked and revised throughout the manuscript.
3. Line 57: what aspect of cognitive performance? Were there any differences in the way different studies measured cognitive functions? My previous comment (Q2) has only partially been addressed.
(The previous comment Q2: Line 49: the authors refer to previous study exploring hypoxia and cognition to have gotten to inconsistent findings, but the reader is left with no details on what these findings were. What did they found? Why is this inconsistent?)
Response: Thank you for your additional comment and we apologize for not providing a satisfactory response in the previous round. In this sentence, we note that, these previous studies using a “similar Go/NoGo task and the same fraction of inspired oxygen, and these studies either reported unaffected or improved cognitive performance in response to exercise.” Thus, the aspect of cognitive performance for all of these studies was an inhibition-based cognitive task, which is assessed from the Go/NoGo task. We then indicate that the inconsistent findings were likely a result of “ … differences in exercise intensity, differences in demographic characteristics, and the timing of the cognitive task.”
4. Lines 64-68: Please note that the change made here in response to my previous comment (see Q3 and Q11) are not satisfactory. Indeed, the points arisen with my previous comments are still unanswered: 1) please define executive control, error processing and inhibition control; 2) what "is likely to be activated when the participants performed the Go/NoGo task"? The subject of this sentence is unclear as there are multiple possibilities in the precedent phrase; and 3) I am still unclear on the reasons that made the authors decide to put so much emphasis on identifying the cortical region associated with inhibitory control, if a deeper investigation of brain activity is not part of the research questions.
(The previous comment Q3: Lines 52-53: “Complex cognitive processes, such as executive control, error processing, and inhibitory control, are partially regulated by the prefrontal cortex (PFC)”. Please define executive control, error processing and inhibition control with references. Also please explain why it is important to specify that these are regulated by the prefrontal cortex – considering that your study is not trying to measure anything related to a specific region of the brain, it might be sufficient to specify what the cognitive processes of interest are.
The previous comment Q11: Line 128: re: “implicating the PFC as an important brain structure involved in cognitively-based inhibitory function”. While this is correct – and clarifies that you are not measuring the overall executive control with the GNG task, contrary to what you have indicated in the introduction – I am still not sure about the reasons that lead you to focus on the PFC. Please explain.)
Response: We have revised the manuscript by defining executive control, error processing and inhibitory control. We have also revised the manuscript by indicating that the prefrontal cortex is the brain structure that is likely to be activated during these types of complex cognitive processes. We feel it is important to mention these complex cognitive processes as well as indicate that the prefrontal cortex is involved in these cognitive processes. We feel this is important because it informs the reader as to why we utilized this cognitive task (Go/NoGo) and what cognitive processes and brain structures are likely to be involved when utilizing this cognitive task.
5. Lines 85-89: Importantly, none of the examples you have provided here are related to executive functions or, more specifically, response inhibition—which is what you measured. Although reference 39 provides some support to your decision to investigate cognitive functions in women, the choice of using GNG is not really supported. Indeed, Sundström Poromaa and Gingnell [39, p.8] reported: "Using tasks that test the ability to inhibit prepotent responses, Colzato et al. reported on less efficient inhibition in the late follicular phase (Colzato et al.,2010), whereas Bannbers and colleagues found no effect of menstrual cycle, either in accuracy or reaction time to a Go-NoGo task (Bannbers et al., 2012). However, using a task that probed inhibitory input control, as opposed to the Stop-signal task and Go-NoGo concerned with inhibitory output control, Colzato and colleagues demonstrated superior inhibition of return in the late follicular phase (Colzato et al., 2012)."
Response: Thank you for your comment. We have removed the examples of visual and spatial processing and have retained the executive functions text. We feel that this is suitable text as the Go/NoGo task measures an aspect of executive function.
6. Line 121: How did you measure the heart rate?
Response: A wireless heart rate monitor was used to measure the heart rate, which was added in Method “Heart rate (HR; Polar F4M BLK, Finland) and…”
7. Lines 155-171: The description of the task is much clearer now. This, however, highlighted that the Go/NoGo task you have used looks more like a Flanker's task than a Go/NoGo task. The Go/NoGo paradigm normally requires participants to inhibit their responses when presented with a NoGo trial (which is the reason why the trial is called NoGo). In your experiment, NoGo trials required participants to press a button. The cognitive skill that allows you to do this is a different aspect of inhibition, known as interference. Also, please note that there is no reference to the cognitive paradigm that you have used. This is clearly not based on the original paradigm developed by Donders (1969). Donders FC. On the speed of mental processes. Acta Psychologica. 1969; 0:412–431. (Original work published 1868)
Response: Thank you for your comment and insights. We have added two references for the cognitive paradigm [4, 5] employed in our study.
4. Ando, S., Hatamoto, Y., Sudo, M., Kiyonaga, A., Tanaka, H., & Higaki, Y., The Effects of Exercise Under Hypoxia on Cognitive Function. PLoS One, 2013. 8(5).
5. Komiyama, T., et al., Cognitive function during exercise under severe hypoxia. Sci Rep, 2017. 7(1): p. 10000.
8. Results: I find the results section a bit confusing and I am afraid that other readers will also have a difficult time with it. Moreover, the collected physiological parameters are not very much integrated with the cognitive data, except from one sudden apparition of SpO2 on line 224-225 which is compared to the study conditions with correlations. I think that you need to clarify much more in the statistical analysis subsection what analysis you have conducted and why, following a logical order. Then, in the result section, the findings should be presented following the same logical order. Adding some lack of clarity, the results related to cognition are presented under the physiolocal parameters subsection.
Response: Agree, and done.
9. Line 225-226: There are some graphical issues with Figure 2. The images appeared cropped in an odd way. Figure 2A presents some issues with the connectors and asterisk over NOR.
Response: Figure 2A has been revised, thank you.
10. Lines 236-238: the sentence seems to say that reaction time decreased indicating that reaction time decreased.
Response: Sorry for the unclear expression, “indicating that the response speed was improved after hypoxic exposure” was deleted.
11. Lines 238-239: this is somewhat conflicting with what you wrote on Lines 45-46: "However, hypoxia itself, may induce negative cognitive-related consequences as severity increases [4, 5].”
Response: Yes, it is. Given that the oxygen availability is reduced during hypoxic exercise, therefore may leading to insufficient oxygen delivery and inadequate cerebral oxygenation and cerebral blood flow, we hypothesized that severe hypoxia used in the present study would impair cognitive performance. However, the actual result was in contrast to our hypothesis.
12. Lines 242-249: I am not sure what you are trying to say with your discussion point on SpO2. Are you trying to explain something related to your study findings regarding cognitive performance?
Response: Thank you for your comment and careful review. We specifically integrated this text on SpO2 because it provides an indirect measure of oxygenation. We observed some differences in cognitive function when compared to other studies. Since SpO2 provides an indirect measure of oxygen, our cognitive function discrepancies with other studies may be a result of differences in cerebral oxygen. We mentioned these SpO2 findings because this could help explain our somewhat discrepant findings.
13. Lines 259-262: Please try to simplify the sentence. It sounds like you are saying the exact opposite of what you have just stated on line 259 (i.e., "be compensated by an increased CBF").
Response: The expression “indicating that cerebral oxygen availability might be diminished during hypoxic exercise” was deleted. Thus the sentence was rephrased as “In the present study, SpO2 was reduced to 87 ± 6% after resting in hypoxia and further decreased to 77 ± 7% after exercise in hypoxia. However, no adverse cognitive consequence was observed, which might be compensated by an increased CBF”.
14. Line 275: What do you mean when you say that SpO2 was not 'enlarged'?
Response: Thank you for your reminder. The statement was revised as “Despite the greater exercise intensity, the reduction of SpO2 in the female subjects of our study was the same as the male subjects in their study [5], both decreasing from 87% to 77%”.
Reviewer 2 Report
The manuscript was adequate revised according to the comments. I have no further comments.
Author Response
Thank you very much!